# The Role of Sex Hormones in Pain-Related Conditions

**DOI:** 10.3390/ijms24031866

**Published:** 2023-01-18

**Authors:** Onella Athnaiel, Santiago Cantillo, Stephania Paredes, Nebojsa Nick Knezevic

**Affiliations:** 1Department of Anesthesiology, Advocate Illinois Masonic Medical Center, Chicago, IL 60657, USA; 2Chicago Medical School, Rosalind Franklin University of Medicine and Science, North Chicago, IL 60064, USA; 3Department of Anesthesiology, University of Illinois, Chicago, IL 60612, USA; 4Department of Surgery, University of Illinois, Chicago, IL 60612, USA

**Keywords:** sex hormones, hormone therapy, pain, pain perception, pain threshold, gender

## Abstract

Millions of people are affected by pain-related conditions worldwide. Literature has consistently shown that each individual experiences and perceives pain in a unique manner due to biological, environmental, and cultural factors in which they have been raised. It has been established that biological males and females perceive pain differently and that it may be partially explained by their distinct hormonal profiles since birth, which are only further magnified during puberty. For biological males, high levels of testosterone have shown to increase their pain threshold; and for biological females, estrogen fluctuations have shown to increase pain intensity and perception. However, sex hormones have not been studied in the context of pain treatment or their impact on biochemical pathways involved in pain perception. For this purpose, the transgender community serves as a unique population to investigate the impact of hormone replacement therapy on molecular pathways involved in the perception of pain. The purpose of this review is to explore the biochemistry of hormone replacement in transgender patients who also have other pain-related conditions such as headaches, fibromyalgia, temporomandibular myalgia, and visceral pain.

## 1. Introduction

The complex interaction between sex hormones and pain has been widely explored by multiple authors, ranging from basic science studies examining the changes in pain neurobiological pathways and pain-related gene expression modification in patients with different hormonal profiles [1], to clinical studies aiming to decipher whether hormonal profiles imply clinically observable differences in pain perception scores and impact on the quality of life [2,3,4].

Although some controversial results have been found, a literature review of published studies implies that estrogens have important effects regulating pain by acting on intracellular receptors, modifying gene expression and G-coupled proteins distributed along the central and peripheral nervous systems [5]. Additionally, estrogens appear to have an influence on other neural pathways related to pain modulation as serotonergic, noradrenergic, dopaminergic and, the more studied, endogenous opioid pathways. In general, estrogens are considered to increase in nociception, and well-described effects have been proved in several pathological conditions such as migraine, fibromyalgia and tension-headache [3,6,7].

The effects of androgens in nociceptive pathways remains unclear; however, the androgen receptor distribution throughout the limbic system and the effects in down-regulation of estrogen receptor expression may be responsible for the accountable clinical differences in pain perception between the two sexes and genders [8,9], considering the role of androgens as being protective against a nociceptive stimulus [7].

Differences between hormonal profiles in pain perception have been studied in the biological females and males, frequently assuming androgens as the main masculine sexual hormone, and estrogens and progesterone as the main female hormones. However, very little data has been published about patients with gender incongruence, when the gender identity is inconsistent with the biological sex, often associated with significant distress, giving rise to Gender Dysphoria (GD) [10], a condition with increasing prevalence requiring a multilateral approach from healthcare providers including use of psychological support and often, hormonal therapy.

In this manuscript, the available literature describing the differences in pain perception and pain-related conditions in biological males and females, the transgender population undergoing sex hormone therapy will be reviewed.

## 2. Epidemiology of Pain in Biological Male/Female and Transgender Patients

Several epidemiologic studies have explored differences in pain perception between males and females, reporting greater prevalence of pain in women, as well as lower pain thresholds [6], increases in chronic pain, and greater prevalence in conditions such as fibromyalgia, migraine, tension headache, irritable bowel syndrome, temporomandibular joint disorders and interstitial cystitis [4]. It has been proposed that, at least in part, and among anatomical and unknown causes, hormonal profiles account for some of the aforementioned differences [4]. There are proven effects of those cholesterol-derived hormones in the modulation of multiple neurobiological pathways related to pain [6], mainly serotonergic^18^ and dopaminergic systems.

Classically, epidemiologic studies describing gender differences in pain perception and modulation have been made in biological male and female patients, assuming that men have higher amounts of androgens and women more estrogens and progesterone [6,9]. However, the transgender population has a different and interesting evolution, behaving to some extent as cross-over patients in which it is possible to trace specific hormonal changes along a timeline, to then evaluate the physiologic evolution aiming at the person’s gender identity, as well as the appearance of undesired adverse effects and pathologies with different prevalence rates between genders. Unfortunately, no epidemiologic descriptions have been published regarding the incidence-related change in transgender patients undergoing cross-sex therapies; however, some authors have described the frequencies of pain-related conditions in small cohorts. Based on physiologic concepts, it is expected that males exposed to long-term estrogen supplementation have a different pain perception and modulation compared with females exposed to prolonged androgen therapy.

## 3. Sex Differences in Immune Response and Pain

Sensation of pain is essential for living organisms to distinguish between safe and dangerous stimuli. There are many triggers for this sensation, namely, noxious thermal, chemical, and mechanical stimuli. In addition to the environmental triggers, tissue damage and inflammation may also lead to pain sensation. The endocrine and immune systems in the human body respond differently to such stimuli; depending on the dominant hormones present, some individuals may experience pain to a higher degree than others [11]. In biological females, estrogens tend to promote a more robust anti-inflammatory response to insults compared to males. However, macrophages, the primary responding cells in the periphery, are more active in the generation of pain in males compared to females [12]. In a consistent manner, microglial cells are highly involved in signaling pain in males compared to females, while they display greater phagocytic features in females compared to males [13]. Peroxisome proliferated-activated receptors alpha and gamma (PPAR-α) and (PPAR-γ) may play a role in relieving pain and display anti-inflammatory features. PPAR-α and PPAR-γ also play a role in the use of fatty acids and storage of lipids, respectively. Interestingly, after a peripheral nerve injury, male mice that were given PPAR-α and female mice that were given PPAR-γ showed reduction in pain, confirming their anti-nociceptive features. Although these immune responses need to be further investigated, literature suggests that the differences in pain perception following stimulation of these receptors may be attributed to sex hormone differences in male and female mice [11].

## 4. Sex Hormones and Pain Perception

Testosterone has consistently shown protection against pain in males. Literature has shown that testosterone limits the production of pro-inflammatory cytokines such as tumor necrosis factor α (TNF-α), which in return contributes to major antinociception in males compared to females^11^. Testosterone decreases capsaicin receptor-mediated signals in the dorsal root ganglion neurons, suggesting that it is antinociceptive [14]. Further, the significantly increased levels of testosterone in biological males may explain the lower prevalence of chronic pain conditions in males compared to females [15].

Literature has shown inconsistent results when considering the effects of estradiol on pain perception in females. Low levels of estrogen have been associated with increased pro-inflammatory response, on the other hand, hormone therapy in biological females has not shown promising results in decreasing the inflammatory response [11]. On the other hand, another study showed that increased pain sensitivity correlated with increased estradiol levels [16]. Although it remains unclear, most studies agree on the impact of estrogen fluctuations on pain perception: fluctuations in hormone levels lead to increased pain, while stable hormone levels serve as a protective mechanism against nociception in females [11,14,17,18]. This is further confirmed by the increased risk of experiencing headaches before the menstrual period because of an abrupt drop in estrogen levels, suggesting the change in hormone levels induces hyperalgesia [19]. Progesterone has also been shown to be involved in pain signaling. Allopregnanolone, a metabolite of progesterone, reduces activation in the trigeminal nucleus caudalis by interacting with GABAA receptors, revealing a protective feature against pain. Progesterone is also believed to act as a neurosteroid in peripheral sensory neurons and in the dorsal horn [14].

Female sex hormones are especially significant during the reproductive age due to the cyclical pattern that they follow monthly. As mentioned earlier, the fluctuating feature of estrogens makes women more vulnerable to a decreased pain threshold, and as a result, more likely to experience severe pain. This is especially interesting when exploring the levels of hormones during the menstrual cycle and its impact on pain sensitivity. One study by Bajaj et al. [20] showed that heat pain threshold was significantly lower during the ovulatory phase compared to the other three menstrual phases (phases are illustrated in Figure 1). Furthermore, the same study showed that the pressure pain threshold is especially decreased in the back during the ovulatory phase. It is worth noting that the ovulatory phase is preceded by elevated estrogen, luteinizing hormone, and follicle stimulating hormone levels and is followed by an immediate drop in those levels. These findings further confirm the hypothesis that the fluctuation of estrogens increases pain sensitivity and increases the likelihood of experiencing pain in the days following a sudden drop in those hormones. The results of this study also confirm the increased sensitivity to stimuli such as heat, pressure, and tactile in biological females compared to males; this highlights the protective features of testosterone in addition to the impact of estrogen fluctuations on the perception of sensory stimuli, specifically those that precipitate pain.

In females at reproductive age that are not receiving oral-contraceptive therapy or other forms of exogenous hormones, pain is typically at the highest intensity following a sudden drop in estradiol, which typically takes place in the middle (days 13–15) of the menstrual cycle [21]. Consistently, within this timeframe, the female population experiences hyperalgesia [19] and lower pain thresholds [20]. Lastly, there is increased pain sensitivity [6] at the peak of estradiol hormone in the follicular phase (days 8–10).

## 5. Common Pain-Related Conditions and Differences between Biological Male/Female and Transgender Patients

Sex hormones play a different role in many of the pain-related conditions that affect biological males and females as well as transgender patients. The following sections attempt to explore the role of hormones on pain perception in different patient populations while considering their hormone profiles. Table 1 provides a summary of the main studies investigating role of sex hormones in pain-related conditions.

### 5.1. Headaches

Migraine headaches and non-migraine headaches are more prevalent in females compared to males [19]. This could be partially explained by the elevated calcitonin-gene related peptides (CGRP) in pregnancy and females receiving oral contraceptive therapy following the estrogen trends in these populations. CGRP is a potent vasodilator and may explain the headaches observed in females. In addition to elevated CGRP, decreased estrogen leads to elevated tryptophan levels, which may exacerbate the headaches in this population as a result of their metabolites such as serotonin and quinolinic acid. Fluctuations of estrogen and their contributions to elevated CGRP and tryptophan confirm the involvement of sex hormones in headaches seen in biological females [14]. The role of testosterone in male headaches remains unclear [19].

Aloisi et al. [22] studied and collected information from 47 transgender females and 26 transgender males, all fulfilling DSM-IV diagnostic criteria for gender identity disorder and who were on long-term estrogen/androgen treatment (for at least one year) to explore sensitivity to sensory stimuli and pain history, including chronic pain. In the female transgender group, 14 patients (29.8%) reported pain on 78% of the occasions with the onset after hormone therapy. Five people reported headaches in more than one location on the head map, sometimes severe enough to cause impairment of daily activities and often exacerbated by environmental stimuli. In two cases the pain was present before hormone therapy but was greatly increased after its onset. In the chronic pain-free group, 18% of the patients reported that they perceived pain earlier and more easily with the symptom lasting longer than usual. Furthermore, all the subjects reported enhanced thermal sensation, while their ability to tolerate the stimuli was decreased. The male transgender group reported a higher prevalence of pain (61%), most commonly headache (13 subjects). In 87% of the subjects, pain was reported as present before hormone therapy. Most of the subjects with cephalalgia marked a single location on the head map; the majority (10/13) reported that it started before hormones, six patients had improvement (less frequency, shorter duration) after testosterone, three had no changes and one experienced increased pain severity. The pain-free group reported no change in nociception or thermal sensitivity.

### 5.2. Temporomandibular Disorders (TMDs)

Temporomandibular joint disorders commonly present with joint pain, crepitus or difficulty chewing and are 1.5–3 times more prevalent in women than in men [23]. Previously described estrogen effects on the temporomandibular joint and nociceptive pathways would likely predict an increased incidence of TMDs in patients undergoing cross-sex hormone therapy. A cyclic pattern in premenopausal women either taking oral contraceptives or not, was proved by LeResche et al. [21] with an increase in pain occurring just before menses in both groups and in the mid-cycle (days 13–15) for the contraception-free group, with a temporal correlation with ovulation and estrogen peak–see Figure 1. In the male group, the pain intensity and characteristics remained unchanged and steady [21]. Interestingly, females that were taking exogenous hormones in the context of oral contraceptive therapy experienced pain to a lesser intensity compared to females that were on a natural cycling of hormones, suggesting the big impact that fluctuations have on pain perception [17].

Likewise, it has been observed that estrogen α-receptor polymorphisms contribute to a different frequency of TMDs in the feminine population in both painful and non-painful presentations [24]. Several other mechanisms have been proposed explaining the link between sex hormones and facial pain, often related to or caused by TMDs [25] symptoms of the disorders usually begin around puberty and peak during the reproductive age [3]. Puri et al. proposed a specific ovarian-estrogen modulator role in neurobiological pathways in the trigeminal ganglion of female mice, with higher levels of neuropeptide Y and galanin mRNA in the estrus phase of the reproductive cycle, when the estrogens exert their greatest influence [26]. A similar study in both female users and non-users of oral contraceptives showed a relative steadiness in pain scores in women taking hormones, different from the cyclic nature of pain in contraceptive non-users [27].

### 5.3. Fibromyalgia

Specifically for fibromyalgia, data suggesting biological reasons accounting for differences in pain presentation and frequencies between genders have been published. The incidence of pediatric fibromyalgia was reported as being similar between age subpopulations until the age of puberty, after which women show a frank increase in the incidence [28]. Schertzinger et al. [29] performed serum level measurements of various hormones in female patients with fibromyalgia correlating with changes in pain presentation during the normal menstrual cycle and found a significant inverse relationship between progesterone and testosterone and pain, and no association with estrogen or cortisol levels. Interestingly, post-hoc analyses evaluating the correlation between cortisol and sex hormones, classifying steroid levels as high, medium, or low, showed that progesterone correlation with pain was only significant at high levels of cortisol, meaning that patients only reported an increase in pain perception when low levels of progesterone and high levels of cortisol coexisted [29]. The result of therapeutic models using testosterone gel for the treatment of fibromyalgia was consistent with the hypothesized role for the hormone, with clinical improvement evidenced in the target population of 11 biological male and female patients [30].

One preclinical study investigating fibromyalgia in Wistar rats showed that reserpine-irreversible vesicular monoamine transporter-2 (VMAT-2)-induced muscle hyperalgesia and allodynia in those rats. Further, female rats that had undergone ovariectomy experienced nociception to a greater extent suggesting that menopause hormone status makes women more susceptible to pain related to fibromyalgia. Ovariectomized female rats experienced 50% reduction in pain perception upon beginning hormone replacement with 17β-estradiol. After 48 h of administration, all groups experienced reduced hyperalgesia and allodynia that was induced by reserpine. These findings suggest that steady levels of estrogens promote less pain perception in females, while sudden drops and fluctuations are major contributors to the increased pain in females with fibromyalgia [18].

As established in many studies, fibromyalgia is more prevalent in biological females than males. This can be partially justified by the fluctuating estrogen and progesterone levels that cycle every month. Interestingly, however, fibromyalgia remains more prevalent in transgender men compared to transgender women. Levit et al. [31] explained that this may be attributed to the exposure of the central nervous system to female hormones during development in-utero. Sex hormones during fetal development may have lifelong effects on pain perception even if patients are receiving hormone therapy during adulthood [31].

### 5.4. Visceral Pain

In an experimental model, rats from both sexes were exposed to mechanical colorectal distention and pain perception was measured. It was observed that female rats responded to less intense painful stimuli in comparison with male rats. However, after ovariectomy, female rats showed a decrease in visceral pain sensitivity at 18 days and orchidectomized rats showed an increase in stress-induced visceral pain. Cross-sex therapy with testosterone and estrogens was administered to intact rats of both genders, after which a decrease in pain duration and intensity was noted in female rats, but no such significant difference was seen in intact male rats. The opposite was also true in male rats undergoing estrogen therapy, showing an increased pain sensitivity [32].

Other experimental models have evaluated an increase in pain thresholds following the administration of testosterone, noticing less responsiveness to nociceptive stimuli after the hormone administration [23,32]. Based on the experimental cross-sex hormone models results and following the epidemiologic trend of other pathologies in both hormonal profiles, it is reasonable to hypothesize that transgender patients might also show a similar behavior for visceral pain.

### 5.5. Musculoskeletal Pain

Musculoskeletal pain is very common in women classified as premenopausal or postmenopausal. This may be attributed to the sudden drop in estrogen upon reaching menopause, which later exacerbates musculoskeletal pain and perhaps precipitates many chronic arthralgias and myalgias [32]. A study by Frange et al. [33] showed that among 510 premenopausal and postmenopausal women, approximately 20% of them reported musculoskeletal pain. Although the study showed no statistical correlation between the reproductive stage and pain, the high percentage of pain perception in this group is clinically significant for physicians treating pain in this population, and their reproductive stage should be taken into consideration [33].

Considering the molecular pathways involved in musculoskeletal pain, preclinical studies have shown that transient receptor potential vanilloid type 1 (TRPV1) is upregulated in biological female rats and in male rats that have received an orchiectomy, which is a procedure that removes one or both testicles. In other words, a lack of or decreased levels of testosterone upregulates TRPV1, which is thought to enhance mechanical sensitivity to pain. This biochemical change is considered to play a major role in pain hypersensitivity in males because testosterone replacement therapy in these rats reduced musculoskeletal pain. However, it did not have a significant effect in females [34].

An interesting finding by Watt showed that females have a protective layer prior to reaching menopause because estrogen seems to block proteinase, which prevents musculoskeletal pain in females. However, this protection is lost following an ovariectomy or upon reaching menopause, which then explains the elevated prevalence of musculoskeletal pain in females and specifically females after menopause [35]. Although estrogen fluctuations in females was considered a trigger for pain and increased pain intensity, perhaps estrogen hormone replacement in these patients may help prevent musculoskeletal pain by providing that protective layer that they previously had prior to menopause.

### 5.6. Breast Pain

Although not one of the most frequently cited pain-related reasons for seeking medical attention, transgender males and females are both at increased risk of breast pain compared to their biological male and female patient cohorts. In transgender males, chest binding, or compressing the chest tissue, is not an uncommon practice [36] and is suspected to promote enhanced mental health [37]. However, recent studies in transgender patients also show an increased frequency of complications, including skin excess and ptosis, abrasions, infection, and pain [37]. Peitzmeier et al. [38] demonstrated a direct correlation between frequency, timing of binding and the usage of certain types of binders (commercial binders, duct tapes, and plastic wraps) with the previously mentioned negative outcomes.

In contrast, transgender females complaining of breast pain are more likely to present with hormone-induced tissue changes, such as gynecomastia. In the Aloisi et al. [22] study, 11 of 14 transgender females reported breast pain, always arising after the start of hormone therapy, in more than 50% unilateral and continuous. This may be partially explained by the pro-nociceptive features of estradiol. Estradiol has shown to increase N-methyl-D-aspartate (NMDA) receptor expression, which is highly involved in pain signaling. In addition, estradiol also leads to increased expression of the TRPV1 and TRPA1 channels, which are commonly associated with endometriosis [39].

### 5.7. Cervical Cancer Screening

It is recommended to biological females to obtain a cervical cancer screening every three years after reaching the age of twenty-one. As a result, this preventative screening is also recommended to transgender men and any patient that has a cervix as the screening can help prevent cervical cancer. Compared to biological females, transgender men tend to experience higher levels of pain during the procedure despite being on testosterone replacement therapy [40]. As established earlier in the article, testosterone was considered a protective hormone against pain as it increased the pain threshold in biological men and in some transgender men that were receiving testosterone replacement therapy. However, it was not effective at removing the pain during this screening procedure. This may be due to many factors, one of which being that cervical screening in transgender men may trigger gender dysphoria and expose the natal anatomy of these patients, and thus making them uncomfortable during the exam. In addition, testosterone also has other effects on the anatomy and the normal functioning of the vaginal canal, which may make the organ more susceptible to pain sensitivity and discomfort during the procedure [41].

## 6. Effect of Pain on Quality of Life

Changes in pain perception and pain-related chronic conditions have been most frequently evaluated as part of the quality-of-life assessment. In the same way, some authors found that chronic pain is an independent quality of life predictor in biological male and female patients [42] as in transgender patients [43]. Between 11–31% of adults suffer from chronic pain worldwide [44]. Considering the heterogeneity of chronic pain and its impact on the quality of life, it is important to understand the protective factors and the risk factors that influence pain perception. Sex hormones play a role in the differences in pain experienced by males and females, and patients receiving hormone therapy. Auer and colleagues [43] evaluated the effects of different conditions on the quality of life, showing chronic pain as an independent predictor of life quality in male transgender patients, without statistical significance in transgender women.

Additionally, besides the sex differences in pain perception, the effect of pain on the quality of life is different between genders, as in the Bingefors et al. [45] study, wherein they demonstrated that headache had more impact on the physical dimensions of quality of life in male patients, while the psychological dimensions were more affected in females. Lastly, certain pain conditions afflict females more than males making this population especially vulnerable for the psychological side effects of pain [44].

As mentioned earlier in the article, cervical screening is especially painful and uncomfortable for transgender men. As expected, this may lead to many transgender men not seeking out preventative care despite it being necessary and beneficial for their wellbeing and longevity. This population, in return, may experience higher rates of cervical cancer and mortality due to late diagnosis and delayed necessary treatments. Educating providers and informing them about the different experiences due to hormone replacement therapy will allow the providers to be more aware and considerate of patient’s unique anatomy, and thus making the experience more comfortable for these patients.

## 7. Conclusions

Literature has consistently shown differences in pain perception in biological males and females. Testosterone shows protective factors in biological males and in transgender males receiving testosterone hormone replacement. On the other hand, estrogen fluctuations provide an explanation for the lower pain threshold seen in females. This is especially true in post-menopausal women who experience a great deal of fluctuations. Very few studies have been conducted evaluating changes in pain perception and pain-related conditions in transgender patients. The correlation of hormonal profiles and the epidemiology of pain-related conditions show a similar distribution of headache and musculoskeletal pain, with feminizing hormone therapy tending to increase pain and testosterone use tending to decrease it. This may be explained by the increased CGRP and tryptophan seen as a result of increased and decreased levels of estrogen, respectively. Similar findings for temporomandibular joint disorders and visceral pain are suggested by animal and therapeutic models. Women suffering from ovarian and breast cancer are especially at risk for experiencing pain due to the sudden increases and decreases in estrogen and progesterone. While the mechanisms of pain perception based on sex hormones have yet to be completely understood, there is no doubt that they influence the intensity of pain in the two sexes and as a result affect the prevalence of chronic pain conditions in these populations.

## Figures and Tables

**Figure 1 ijms-24-01866-f001:**
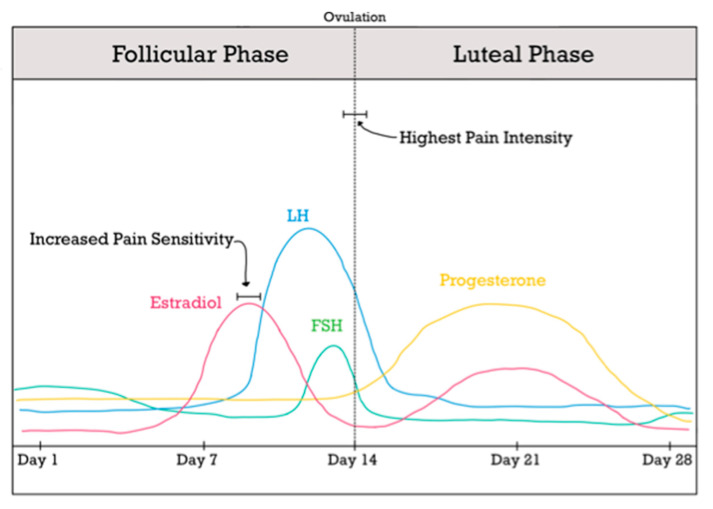
Summary of Female Sex Hormones During the Menstrual Cycle.

**Table 1 ijms-24-01866-t001:** Sex Hormones in Pain-Related Conditions.

Pain-Related Condition	Major Biochemical Pathways	Main Findings
Headaches	Estrogen CGRP	Biological females are more likely to experience headaches due to the impact of estrogen on the potent vasodilator, CGRP. This is especially prevalent during pregnancy and in biological females receiving estrogen oral contraceptive pills. Male transgender patients receiving hormone replacement therapy experienced more pain than their female transgender counterparts highlighting the impact of sex hormones on pain perception.
Temporomandibular Disorders	Estrogen Neuropeptide Y Galanin mRNA	Biological females are more likely to experience temporomandibular pain prior to estrogen peak and ovulation. Females receiving steady estrogen replacement through oral contraceptives, were less likely to experience pain. Elevated neuropeptide Y and galanin mRNA are also commonly observed, contributing to the impact of estrogen on pain perception.
Fibromyalgia	VMAT-2 Estrogen	Postmenopausal women are more likely to experience pain related to fibromyalgia. Estrogen replacement reduces the pain by 50%, confirming the benefit of steady estrogen levels as opposed to the cyclical pattern. In-utero exposure to female hormones places transgender men at higher risk of fibromyalgia than transgender women patients despite hormone replacement therapy.
Visceral Pain	Estrogen Androgens Testosterone	Lack of estrogen increased stress-induced pain sensitivity while testosterone administration reduced the pain sensitivity, confirming the protective features of male sex hormones. Female hormone replacement increased pain sensitivity suggesting similar results in transgender women patients.
Musculoskeletal Pain	TRPV1 Testosterone	Increased mechanical sensitivity is seen in biological female rats and male rats that are not producing the appropriate testosterone levels. Pain is up regulated by the presence of TRPV1 receptors. Testosterone replacement reduced the pain in males but did not influence the female rats.
Female Reproductive Organs	Estrogen NMDA TRPA1 TRPV1	Estrogen upregulated the expression of NMDA, TRPA1, and TRPV1, and all of which play a role in increasing pain perception and enhancing molecular signaling of pain. Transgender men and women are at risk for breast pain. In transgender men, the physical binding increases their pain levels while in transgender women, the hormone replacement therapy seems to precipitate the pain most likely due to tissue changes.

## Data Availability

Not applicable.

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
