# Peer review of "The Role of Sex Hormones in Pain-Related Conditions"

_ijms, 2023, doi:10.3390/ijms24031866_

Round 1

Reviewer 1 Report

The manuscript summarizes the different effects of gender and sex hormones, such as testosterone and estrogen, on the pain process. In addition, the authors note that the transgender population is a unique group that receives different types of sex hormone replacement therapy and therefore may have different sensitivities and adjustments to various types of pain, including headache, fibromyalgia, temporomandibular myalgia and visceral pain. The investigation sheds new light on the role of gender and sex hormones in the development of pain. However, I have a few questions for the authors to address before this manuscript is ready for publication.

Questions:

  1. I suggest that the authors add to Figure 1 the effect of different female sex hormone levels and cycles on pain, rather than just showing the basics of how hormone levels change during ovulation.
  2. The authors describe "most studies agree on the impact of estrogen fluctuations on pain perception" on page 3 of the manuscript, but only one literature is cited.
  3. Is Table 1 given by the authors a summary of section 5? If yes, please indicate this in the manuscript.
  4. The main topic of this review is about the difference in pain between biological male/female and transgender patients. However, Table 1 given by the author only briefly describes the relationship between different pain and sex hormones, CGRP, etc. Therefore, it is recommended that the authors add the regulation of sex hormones on pain in transgender patients to Table 1.

Author Response

1. Reviewer’s comment: “The manuscript summarizes the different effects of gender and sex hormones, such as testosterone and estrogen, on the pain process. In addition, the authors note that the transgender population is a unique group that receives different types of sex hormone replacement therapy and therefore may have different sensitivities and adjustments to various types of pain, including headache, fibromyalgia, temporomandibular myalgia and visceral pain. The investigation sheds new light on the role of gender and sex hormones in the development of pain. However, I have a few questions for the authors to address before this manuscript is ready for publication.”

Response: Thank you very much for your feedback, we are happy to revise and answer any questions about the manuscript.

2. Reviewer’s comment: “I suggest that the authors add to Figure 1 the effect of different female sex hormone levels and cycles on pain, rather than just showing the basics of how hormone levels change during ovulation.”

Response: Thank you for your suggestion. We have updated the figure to highlight the timing of highest pain intensity and increased pain sensitivity. Additionally, we have added a caption beneath the figure explaining patterns of increased pain intensity and the population in which it is most commonly seen.

3. Reviewer’s comment: “The authors describe "most studies agree on the impact of estrogen fluctuations on pain perception" on page 3 of the manuscript, but only one literature is cited.”

Response: Thank you for this feedback. We have reviewed the other articles and have added more references to support this sentence.

4. Reviewer’s comment: “Is Table 1 given by the authors a summary of section 5? If yes, please indicate this in the manuscript.”

Response: Thank you for the suggestion. We have added a sentence at the beginning of section 5 pointing to Table 1.

5. Reviewer’s comment: “The main topic of this review is about the difference in pain between biological male/female and transgender patients. However, Table 1 given by the author only briefly describes the relationship between different pain and sex hormones, CGRP, etc. Therefore, it is recommended that the authors add the regulation of sex hormones on pain in transgender patients to Table 1.”

Response: Thank you for your recommendation. Additional information was added to Table 1 to highlight the role of sex hormones on pain in transgender patients.

Reviewer 2 Report

I appreciate the author presenting this review article. This is a well-reviewed and organized article. This summarized and nice information may provide to the readers.

Author Response

  1. "Reviewer’s comment: “I appreciate the author presenting this review article. This is a well-reviewed and organized article. This summarized and nice information may provide to the readers.”

Response: Thank you for the positive feedback, your evaluation is greatly appreciated.

Reviewer 3 Report

This is a very interesting paper regarding the role of sexual hormones in different pain conditions.

          Estrogens increase nociception. They act on pathways involved in pain modulation, on intracellular receptors, modifying gene expression and G-coupled proteins distributed in central and peripheral nervous system. Also estrogens acts on serotonergic, noradrenergic, dopaminergic and endogenous opioid pathways. In biological females, the fluctuations in estrogen levels increase pain intensity and perception. In females at reproductive age the fluctuations in hormone levels determine increased pain (for example headache before he menstrual period because of an abrupt drop in estrogen levels); while stable hormone levels are protective mechanism against nociception.

          Androgens are protective against nociceptive stimulus. Testosterone receptors are distributed in limbic system. In biological males, high levels of testosterone increase their pain threshold.

          There is also a greater prevalence of chronic pain conditions in women (migraine, tension headache, fibromyalgia, temporo-mandibular joint dysfunction).

Estrogens tend to promote a more robust anti-inflammatory response to insults, while testosterone reduce the production of pro-inflammatory cytokines tumor necrosis factor É‘.

          There are very few studies regarding the role of sex hormones replacing therapies in pain conditions in transgender people. We suppose that in males exposed to long-term estrogen supplementation may be a different pain perception compared with females exposed to prolonged androgen therapy. In cross-sex therapies - feminizing hormone therapy tending to increase pain and testosterone use tending to decrease it. In transgender man the screening for cervical cancer is a particularly painful procedure, they tend to avoid it.

Author Response

  1. Reviewer’s comment: “This is a very interesting paper regarding the role of sexual hormones in different pain conditions. Estrogens increase nociception. They act on pathways involved in pain modulation, on intracellular receptors, modifying gene expression and G-coupled proteins distributed in central and peripheral nervous system. Also estrogens acts on serotonergic, noradrenergic, dopaminergic and endogenous opioid pathways. In biological females, the fluctuations in estrogen levels increase pain intensity and perception. In females at reproductive age the fluctuations in hormone levels determine increased pain (for example headache before the menstrual period because of an abrupt drop in estrogen levels); while stable hormone levels are protective mechanism against nociception.

Androgens are protective against nociceptive stimulus. Testosterone receptors are distributed in limbic system. In biological males, high levels of testosterone increase their pain threshold.

There is also a greater prevalence of chronic pain conditions in women (migraine, tension headache, fibromyalgia, temporo-mandibular joint dysfunction).

Estrogens tend to promote a more robust anti-inflammatory response to insults, while testosterone reduce the production of pro-inflammatory cytokines tumor necrosis factor É‘.

There are very few studies regarding the role of sex hormones replacing therapies in pain conditions in transgender people. We suppose that in males exposed to long-term estrogen supplementation may be a different pain perception compared with females exposed to prolonged androgen therapy. In cross-sex therapies - feminizing hormone therapy tending to increase pain and testosterone use tending to decrease it. In transgender men the screening for cervical cancer is a particularly painful procedure, they tend to avoid it.”

Response: Thank you for the positive feedback, we appreciate your evaluation and your emphasis on the key points of our manuscript.

Round 2

Reviewer 1 Report

The authors have addressed all my concerns in a satisfactory way. I have no further questions.